# Transient Inflammation of Pancreatic Exocrine Tissue in Autoimmune Diabetes Follows Onset of Islet Damage and Utilizes Heparanase-1

**DOI:** 10.3390/ijms26094120

**Published:** 2025-04-26

**Authors:** Charmaine J. Simeonovic, Zuopeng Wu, Sarah K. Popp, Gerard F. Hoyne, Christopher R. Parish

**Affiliations:** 1Immunology and Infectious Diseases Division, The John Curtin School of Medical Research, The Australian National University, Canberra, ACT 0200, Australia; 2Genome Sciences and Cancer Division, The John Curtin School of Medical Research, The Australian National University, Canberra, ACT 0200, Australia; zuopeng.wu@health.gov.au (Z.W.); sarah.popp@anu.edu.au (S.K.P.); christopher.parish@anu.edu.au (C.R.P.); 3School of Health Sciences, The University of Notre Dame Australia, Fremantle, WA 6160, Australia; gerard.hoyne@nd.edu.au

**Keywords:** exocrine pancreas, inflammation, autoimmune diabetes, heparanase-1 (HPSE-1), islet

## Abstract

Inflammation of the exocrine pancreas accompanies autoimmune diabetes in mouse models and humans. However, the relationship between inflammation in the exocrine and endocrine (islet) compartments has not been explored. To address this issue, we used a transgenic mouse model in which autoimmune diabetes is acutely induced after the transfer of islet beta cell-specific transgenic T cells. Histological analyses demonstrated that inflammation of the exocrine pancreas, which was initially mild, resulted in the transient but widespread disruption of acinar tissue. Islet inflammation preceded exacerbated exocrine pathology, progressed to T cell-induced islet damage/destruction and persisted when exocrine inflammation subsided. Heparanase-1 (HPSE-1), an endoglycosidase that degrades heparan sulfate in basement membranes (BMs), when preferentially expressed in recipient cells but not donor (HPSE-1-deficient (HPSE-KO)) T cells, played a critical role in both exocrine and islet inflammation. In this context, HPSE-1 facilitates the passage of autoimmune T cells across the sub-endothelial basement membrane (BM) of pancreatic blood vessels and initially into the exocrine tissue. Peak exocrine inflammation that preceded or accompanied the acute onset of diabetes and HPSE-1 potentially contributed to acinar damage. In contrast to inflammation, HPSE-1 expressed by donor T cells played a key role in the induction of diabetes by allowing autoimmune T cells to traverse peri-islet BMs in order to destroy insulin-producing beta cells. Overall, our findings suggest that major exocrine pancreas injury is not required for the initiation of autoimmune islet damage and is not essential at the time of diabetes onset.

## 1. Introduction

Type 1 diabetes (T1D) is typically defined as a chronic autoimmune disease that destroys the insulin-producing beta cells in the Islets of Langerhans (the endocrine compartment) in the pancreas. However, growing evidence supports the concept that the disease process is heterogenous and can also involve inflammation of the exocrine pancreas. The exocrine compartment consists of acinar tissue that represents the vast majority of the pancreas (>95% by mass) and produces and secretes an array of enzymes to aid digestion, e.g., trypsinogen, chymotrypsinogen, elastase, amylase, and lipase [1]. Studies of human T1D have previously revealed moderate levels of exocrine dysfunction in fecal elastase-1, serum amylase, and lipase [2,3,4]. Decreased serum P-amylase and lipase were also found in adults with a high genetic risk of developing T1D [5]. Transcriptome analyses of pancreas tissue from patients with recent onset T1D revealed gene expression changes in both beta cells and acinar cells, confirming the co-involvement of the exocrine compartment in T1D progression [6]. In further support, Magnetic Resonance Imaging (MRI) also demonstrated a significant decrease in pancreas volume during the first year after T1D diagnosis [7]. In the context of inflammation, the immunofluorescence staining of pancreata from human donors with established T1D have shown an increased density of CD8 T cells and CD4 T cells in exocrine tissue, a finding that paralleled a significant decrease in pancreas weight [8]. Inflammatory cells, including innate cells of the immune system, also were reported in the exocrine pancreas of non-obese diabetic (NOD) mice [9,10]. Nevertheless, the mouse models of T1D have largely been overlooked for investigating the relationship between exocrine tissue inflammation and autoimmune islet damage during T1D development, despite the logistical difficulties in characterizing these processes in human T1D [1].

We have previously utilized an adoptive transfer model of T1D induction in transgenic RIP-OVA^hi^ mice, which expressed ovalbumin (OVA) in their islet beta cells, to investigate the requirements for T cell migration, islet inflammation, and the onset of T1D [11]. We reported that adoptively transferred OVA-specific transgenic OTI (CD8) T cells and OTII (CD4) T cells, as well as other host cells, co-operatively produce the endoglycosidase heparanase-1 (HPSE-1) to enable migration of the OT T cells to host islets, the subsequent damage of OVA-expressing beta cells, and the acute induction of T1D within 2 weeks after cell transfer [12]. Heparanase, produced by multiple cell types, e.g., T cells, endothelial cells, and platelets, therefore plays a critical role in promoting the inflammation that induces autoimmune T1D, a property that highlights the potential for heparanase inhibitors as therapeutic anti-inflammatory agents [13,14,15,16]. Using pancreas samples from our earlier study in RIP-OVA^hi^ mice [12], we carried out a retrospective histological analysis of exocrine pancreas inflammation and islet inflammation/infiltration at 3 to 14 days after cell transfer, including the time of T1D onset. Our findings highlight major differences in the kinetics and persistence of inflammation in the exocrine versus endocrine (islet) tissue compartments during the development of T1D and in the respective requirements for HPSE-1 by donor OT T cells and host (e.g., non-T) cells.

## 2. Results

### 2.1. Transfer of Islet Antigen-Specific T Cells Induces Acute Inflammation in the Exocrine Pancreas of RIP-OVA^hi^ Mice

In previous studies, the transfer of naïve OVA-specific OTI T cells and activated OVA-specific OTII T cells in RIP-OVA^hi^ mice focused solely on islet-mediated damage and the subsequent induction of diabetes [11,12]. Using this acute model for inducing diabetes, we confirmed our previous discovery of a major role for the heparan sulfate (HS)-degrading enzyme HPSE-1 in islet inflammation and immune-mediated damage of insulin-producing beta cells [12,16]. We have now examined whether this same transfer of antigen-specific T cells in RIP-OVA^hi^ mice represents a suitable and convenient acute model for studying the relationship between the inflammation of pancreatic exocrine (acinar) tissue and endocrine (islet) tissue during diabetes development.

Firstly, we investigated whether the inflammation of pancreatic exocrine tissue occurs after the transfer of activated OTII and naïve OTI T cells to RIP-OVA^hi^ mice. Host pancreata at 3–14 days after cell transfer were examined histologically, and the level of immune cell infiltration into acinar tissue (AT) and islets (I) was assessed using an arbitrary scoring system. Inflammation was low-grade or absent in both tissue compartments in 83.3% of pancreata at day 3 post-cell transfer, with the remaining 16.7% of pancreata preferentially demonstrating more pronounced islet infiltration compared to acinar tissue (Figure 1 and Figure 2a,b). Against a consistent background of a light cellular infiltrate (LI) in exocrine tissue on day 6, one-third of the samples demonstrated peri-islet (Score 2), mild intra-islet (Score 3), and pronounced (Score 4) islet infiltration by immune cells and hence progression of the autoimmune response leading to damaged islets (DI; Figure 1). Pancreata harvested at day 9 showed a significant increase in intra-pancreatic inflammation compared to RIP-OVA^hi^ mice without OT T cell transfer (*p* = 0.04, Score 0); 66.7% of host pancreata demonstrated gross, widespread infiltration (WI) and damage of acinar tissue (DA) and islets (Figure 1 and Figure 2c). The remaining 33.3% of pancreata showed preferential immune-mediated damage of islets compared to acinar tissue. Pancreatic inflammation at days 10–12 was marked predominantly by islet damage and only mild exocrine tissue infiltration (MI; Figure 1 and Figure 2d). By day 14, pancreatic inflammation appeared to decline in the presence of islet destruction (ID; Figure 1 and Figure 2e). The RIP-OVA^hi^ experimental model of diabetes induction therefore demonstrated inflammation of the exocrine pancreas and transient widespread acinar tissue disruption and destruction but pronounced and sustained immune-mediated islet damage. Importantly, evidence of early marked immune-mediated damage to islets preceded the exacerbation of immune cell infiltration into acinar tissue.

### 2.2. Properties of Donor Islet Antigen-Specific T Cells That Induce Pancreas Inflammation and Diabetes in RIP-OVA^hi^ Mice

All RIP-OVA^hi^ mice that received 2 × 10^6^ naïve OTI cells combined with 2 × 10^6^ activated OTII cells were diabetic on days 10–12 after cell transfer (*n* = 4/group; Figure 3a). Maximum diabetes incidence (100%) correlated with moderate to heavy islet infiltration but only mild inflammation of the exocrine pancreas component (Score 4, *p* = 0.03 for T1D versus non-T1D) (Figure 3a,b). However, this direct correlation was not evident by day 14 after transfer of the same dose of naïve OTI T cells, activated OTI cells, or increased numbers of naïve OTI T cells. The lower incidence of diabetes (37.5%) at 14 days after the transfer of naïve OTI cells and activated OTII cells, compared to 10–12 days, correlated with a decline in only islet inflammation (37.5%, Scores 0–2; *n* = 8/group). This finding was also consistent with the zero incidence of diabetes and high level of low-grade islet inflammation (40%, Score 3) after the transfer of 2 × 10^6^ activated OTI cells (*n* = 5/group). Interestingly, diabetes onset in 25% of host mice (*n* = 4/group) at 14 days after transfer of 4–6 × 10^6^ naïve OTI cells correlated directly with peak inflammation of both pancreatic acinar and islet tissue compartments. Taken together, the data in Figure 1 and Figure 3 suggest that maximum exocrine and islet tissue inflammation can precede or accompany the acute onset of diabetes.

### 2.3. Destructive Exocrine Tissue Inflammation in RIP-OVA^hi^ Mice Preferentially Utilizes Recipient HPSE-1

We previously reported that HPSE-1 expression in both donor OT T cells and recipient RIP-OVA^hi^ mice significantly contributes to the development of diabetes [12]. In this study, we investigated whether HPSE-1 is essential for the accompanying inflammation in pancreatic exocrine tissue. Based on the optimal cell dose required for acinar and islet tissue inflammation as well as diabetes induction previously identified (Figure 1 and Figure 3), these molecular studies utilized the adoptive transfer of 2 × 10^6^ naïve wildtype (WT) or HPSE-1 deficient OTI cells and activated OTI cells to WT or HPSE-1 deficient RIP-OVA^hi^ mice.

Significant exocrine tissue inflammation accompanied by damage to acinar tissue, as well as islets, occurred as early as day 6 and persisted up to day 9 after the transfer of HPSE-1-deficient donor cells (*p* = 0.01, Score 5, 6 days versus 3 days; *p* = 0.002, Score 5, 9 days versus 3 days) (Figure 4). In contrast, increased exocrine inflammation was observed only at day 9 after the transfer of WT OT T cells (*p* = 0.04, Score 0, 9 days versus WT), with 66.7% of host pancreata demonstrating gross immune cell infiltration throughout the acinar tissue compartment and islets (Score 5; Figure 4). These findings indicate that HPSE-1 expression in recipient cells rather than donor T cells plays a critical role in both exocrine tissue inflammation and disruption, as well as in the immune damage of islets.

In further support, HPSE-1 deficiency in RIP-OVA^hi^ mice led to the major infiltration of acinar tissue as well as acinar and islet destruction after the transfer of WT cells only at day 9 (*p* = 0.02, Score 5, 9 days versus 3 days) (Figure 4). Unlike WT hosts receiving HPSE-1-deficient donor cells, no significant increase in exocrine and endocrine tissue pathology was observed between day 3 and day 6. These findings suggest that, despite the expression of HPSE-1 in donor T cells, acinar and islet tissue inflammation proceeded at a slower tempo and hence less efficiently in the absence of host HPSE-1. Recipient HPSE-1 therefore contributes to exocrine tissue inflammation as well as to islet inflammation. Nevertheless, the absence of both host and donor HPSE-1 did not prevent major pancreatic inflammation (at 6–14 days).

Our data further suggest that intra-islet infiltration and islet damage (Score 4) is an early event, correlating with HPSE-1 expression in WT donor cells as early as day 3 (WT recipient) and day 6 (WT and HPSE-1-deficient recipients) and normally precedes damaging exocrine pancreas inflammation (day 9) (see Section 2.4). Altogether, our findings suggest that exocrine pancreas injury is not a prerequisite for the initiation or persistence of antigen-specific islet damage.

### 2.4. Donor HPSE-1 Plays a Dominant Role in the Induction of Diabetes in RIP-OVA^hi^ Mice

After identifying a major role for recipient HPSE-1 in gross pancreas inflammation and particularly exocrine tissue infiltration and destruction at 6–9 days after OT T cell transfer to WT RIP-OVA^hi^ mice (see Figure 4), and, given that diabetes onset generally peaks at 10–11 days [12], we investigated the relationship between the expression of donor or recipient HPSE-1, exocrine tissue inflammation, and diabetes induction at the later time of 10–14 days after transfer.

The transfer of WT OT T cells to WT RIP-OVA^hi^ mice induced diabetes in all recipient mice (100%, *n* = 4/group; Figure 5a) at 10–12 days, correlating with the light infiltration of exocrine pancreas and islet infiltration and destruction (Score 4) (Figure 6a). However, the induction of diabetes was also observed in 100% of HPSE-1-deficient RIP-OVA^hi^ recipient mice that received WT OT T cells, despite a significant difference in exocrine tissue histology between WT and HPSE-1-deficient RIP-OVA^hi^ mice (*p* = 0.008, Score 5, WT versus HPSE-1-deficient hosts) (Figure 5b and Figure 6b). By 14 days after the transfer of WT OT T cells, the incidence of diabetes onset in WT mice decreased to 40%, compared to 100% at 10–12 days (Figure 5a). Furthermore, no significant differences were observed in diabetes incidence or exocrine histology in WT mice that received HPSE-deficient OT T cells (Figure 5a,b), despite 50% of pancreata showing major acinar tissue disruption (Score 5) compared to 0% after the transfer of WT cells (62.5%, Score 4). These findings strongly suggest that HPSE-1-expressing WT donor cells play a critical role in the induction of diabetes by 10–12 days and that gross exocrine tissue inflammation (Score 5) is not essential at the time of diabetes onset.

## 3. Discussion

Clinical studies of exocrine tissue inflammation in pancreata from human donors with recent or established T1D hereto have failed to resolve the question of whether exocrine tissue inflammation precedes, accompanies, or follows the autoimmune-mediated assault of beta cells in pancreatic islets. We report here that, like human T1D, inflammation of pancreatic exocrine tissue accompanies the acute development of diabetes induced by the adoptive transfer of OT T cells in RIP-OVA^hi^ mice. Semi-quantitative histological analyses of pancreata at different time points (3–14 days) revealed an initial focal, low-level immune cell infiltrate within the acinar tissue compartment up to and including day 6, possibly suggesting the passage of autoimmune T cells to islets and intra-islet beta cells. Indeed, using intra-vital two-photon microscopy for tracking beta-cell-specific T cells in other transgenic models of diabetes, immune T cells have been shown to extravasate from pancreatic exocrine blood vessels and initially migrate through exocrine tissue [17,18]. Also, by day 6 in our study, the progression of islet inflammation and autoimmune damage of islet tissue was evident, in contrast to the minor infiltration in exocrine tissue. By the later time of day 9, widespread infiltration and the injury of both acinar tissue and islets were observed, with only islet tissue continuing to show evidence of significant immune damage thereafter (days 10–14; Figure 1). While peak pancreatic exocrine tissue inflammation/damage coincided with that of endocrine islet tissue, the development of inflammation in acinar tissue progressed slower and was more transient than for islets. These outcomes support a more prominent role for antigen-specific T cell responses in beta cell destruction and diabetes development in the RIP-OVA^hi^ model. Our study further demonstrates that exacerbated acinar tissue inflammation and damage can precede or accompany diabetes onset and hence is not essential at this point in disease progression. This finding suggests that exocrine inflammation may represent an intermediate migratory stage for beta-cell specific T cells [17,18] as well as host innate cells (e.g., macrophages [12], neutrophils [19,20]), sometimes resulting in collateral damage to pancreatic acini. However, this does not exclude a possible role for acinar debris indirectly amplifying the non-antigen specific inflammation of islets by innate immune cells. In addition to T cell-mediated beta cell death [12], the release of supplementary damaging mechanisms by innate immune cells [19,20] could also contribute to beta cell injury and diabetes pathogenesis.

The efficient passage of immune cells from blood vessels into underlying parenchymal tissue requires the deployment of matrix-degrading enzymes [15,21,22]. Heparanase-1 plays an important role in this process by cleaving the long side chains of the largest heparan-sulfate proteoglycan perlecan present in the sub-vascular basement membrane (BM). This process weakens the BM structure, providing exit pathways to facilitate leukocyte migration into tissues [15,21]. Using RIP-OVA^hi^ bone marrow chimeras, we previously reported that HPSE-1 from multiple sources, including activated donor OTII (CD4) T cells, and host cells (e.g., radio-resistant endothelial cells and bone marrow-derived platelets) are required for the optimal induction of diabetes [12]. However, our current histological studies of compartment-specific inflammation in the pancreas have demonstrated that host HPSE-1 plays a more dominant role than donor T cell-derived HPSE-1 in generating both exocrine and endocrine tissue inflammation in RIP-OVA^hi^ mice. We speculate here that host endothelial HPSE-1 is critical for the passage of immune cells from the vasculature into the pancreas, with subsequent leukocyte migration resulting in the inflammation of both exocrine tissue and islets.

In addition to the sub-vascular BM, pancreatic acini and islets are also surrounded by their own individual acinar and peri-islet BMs, respectively, that also contain the HSPG perlecan [23,24]. Furthermore, immunohistochemical studies have demonstrated HPSE-1-expressing islet-infiltrating leukocytes [16,25] and nearby acinar-infiltrating immune cells [25] in pancreata from NOD mice and humans with recent-onset T1D. Our data from the RIP-OVA^hi^ model reported here demonstrate an important role for donor T cell-derived HPSE-1 in the induction of diabetes. We reason that HPSE-1 produced locally by donor T cells that have already migrated to the islet perimeter remodels and relaxes the islet BM scaffold, thereby allowing the entry of antigen-specific T cells into the islets, beta cell damage and destruction, and ultimately the initiation of diabetes. Likewise, we suggest that the local production of HPSE-1 by host innate immune cells, e.g., neutrophils and platelets [26,27], that can accompany T cells during diabetes pathogenesis [20,28] could disrupt the BM of acini, possibly leading to the apoptosis of acinar cells via anoikis [29]. Importantly, we have found that, during T1D development in both NOD mice and humans, islet-infiltrating immune cells use HPSE-1 to degrade intracellular heparan sulfate (HS) localized in islet beta cells, eliminating an important protective mechanism for beta cells against reactive oxygen species (ROS)-mediated cell damage and death [16,25]. During diabetes development, HPSE-1 can therefore function by aiding leukocyte migration into the pancreas, establishing exocrine and islet inflammation, islet infiltration, and beta cell damage, as well as potentially acinar cell death.

In summary, our molecular studies using the RIP-OVA^hi^ mouse model of diabetes induction emphasize that the cellular source of HPSE-1 is site-specific. Inflammation of both the exocrine and endocrine pancreas compartments and potentially acinar tissue injury preferentially use host-derived HPSE-1 (e.g., from endothelial cells and innate immune cells). In contrast, HPSE-1 deployed mainly by donor diabetogenic T cells results in islet entry, beta cell destruction (due directly to HPSE-1 as well as other damaging immune mechanisms), and ultimately in diabetes onset. Nevertheless, our finding that a lack of both donor and host HPSE-1 did not prevent exocrine and islet inflammation is not surprising. Other matrix-degrading enzymes, such as proteases, e.g., cathepsins C, S, and W, have also been reported to contribute to leukocyte ingress into NOD mouse islets and islets in pancreata from human donors with T1D [24]. Thus, certain proteases may also aid leukocytes to traverse the sub-vascular BM and thereafter damage acini BMs and disrupt the exocrine pancreas.

The RIP-OVA^hi^ model represents an experimental system for inducing diabetes by the adoptive transfer of beta cell-specific OT T cells and therefore differs from the spontaneous development of autoimmune T1D in humans and NOD mice. Nevertheless, our study has revealed an important role for HPSE-1 in pancreatic exocrine inflammation during diabetes development. Significantly, our findings also indicate that the major inflammation of pancreatic acinar tissue and exocrine tissue damage is not required in the early stages of the initiation and progression of islet inflammation, co-exists with advanced islet inflammation in the pre-diabetes stage but does not necessarily accompany diabetes onset. The progression of exocrine tissue inflammation, albeit transient, may, however, embellish the immune response to islet beta cells in an antigen non-specific manner. Future studies of human pancreata from autoantibody-positive and recent-onset donors will be required to ascertain whether these outcomes are relevant for human T1D. In addition, heparanase inhibitors that have previously been shown to protect against islet inflammation, beta cell demise, and T1D development [16,25,30] may also be useful therapeutics for reducing pancreatic exocrine tissue inflammation.

## 4. Materials and Methods

### 4.1. Mice

C57BL/6J OT-I, OT-II, and RIP-OVA^hi^ mice [11,31] were obtained from the Walter and Eliza Hall Institute, Australia. HPSE-1-deficient C57BL/6J mice were kindly provided by Dr I Vlodavsky and Dr J Li [32]. All mice were bred and maintained in the specific pathogen-free Australian Phenomics Facility (Australian National University (ANU)), including cross-breeding to produce HPSE-1 deficient strains and CD45.1-positive strains, as previously reported [12]. Only male donor and recipient mice were used for transfer experiments [12]. All animal procedures were approved by the Australian National University Animal Experimentation Ethics Committee (AEEC).

### 4.2. Diabetes Induction

Lymph nodes from male OT I and OT II mice were used as a source of OT T cells for activation in vitro (only OTII cells) and adoptive transfer to RIP-OVA^hi^ mice [11,12]. For activation of OTII T cells, suspensions of lymph node cells were stimulated with irradiated C57BL/6J spleen cells in the presence of OVA_323–339_ peptide (1 µg/mL and subsequently 0.5 µg/mL; Biomolecular Resource Facility, JCSMR, ANU) [33], rIL-2 (10 ng/mL), and LPS (5 µg) [12,34]. Naïve OT-I and activated OT-II cells were purified by negative selection using a cocktail of hybridoma supernatants (M5/114, Ter119, M1/70, F4/80, and RB6.8C5) together with GK1.5 mAb for OT-I cells and 53.6.7 mAb for OT-II cells, followed by BioMag^®^ Goat anti-Rat IgG (Qiagen, Hilden, Germany) and magnetic separation [12]. CD45.1 OT T cells were able to be distinguished from CD45.2-positive C57BL/6J cells during assessment of donor cell purification by flow cytometry. For the majority of experiments, RIP-OVA^hi^ mice were injected iv with 2 × 10^6^ OT-I and 2 × 10^6^ OT-II cells in 200 µL [12]. Blood glucose was measured daily for up to 14 days, using a glucometer. Mice registering 2 consecutive hyperglycemic readings (>13 mmol/L) were identified as diabetic and were then sacrificed [12].

### 4.3. Flow Cytometry

Data to assess OT cell purification were acquired on a BD LSR II and analyzed using FlowJo software (version 10.0.0, TreeStar, Becton Dickinson, Franklin Lakes, NJ, USA). For some experiments designed to study cell migration post-transfer and previously reported [12], OT T cells were labeled with Carboxyfluorescein succinimidyl ester (CFSE) or Cell Trace Violet (CTV) (Molecular Probes, Eugene, OR, USA). In this study, experiments using such cells were used to investigate time to diabetes induction after cell transfer and are identified as “labeled” in the relevant figure legend.

### 4.4. Histology and Assessment of Pancreas Inflammation

Pancreata from RIP-OVA^hi^ mice and HPSE-deficient RIP-OVA^hi^ mice post-adoptive transfer of OT T cells or without cell transfer were fixed in 10% neutral-buffered formalin, and paraffin sections (4 µm thick) were stained with hematoxylin and eosin (H&E), as previously reported [12]. Coded (unidentified) H&E-stained pancreas sections were subjected to blinded examination and were scored using a Laborlux 12 light microscope (Leitz, Wetzlar, Germany). The severity of leukocyte inflammation was assessed in a representative section of each pancreas using an arbitrary scoring system: Score 0—no peri-islet leukocytes and little/no exocrine pancreas infiltration; Score 1—some peri-islet leukocytes and little/no exocrine pancreas infiltration; Score 2—peri-islet leukocytes and patches of light/moderate infiltration in exocrine pancreas; Score 3—some islet infiltration and patches of light/moderate infiltration in exocrine pancreas; Score 4—some moderate/heavy islet infiltration and patches of light/moderate infiltration in exocrine pancreas; and Score 5—moderate/heavy islet infiltration/damage and areas of widespread exocrine pancreas infiltration with gross acinar tissue damage. At the completion of histological scoring, the samples were decoded. The % of pancreata with each score was determined for each group. Representative pancreas sections were photographed using an AxioObserver inverted microscope (Zeiss, Jena, Germany).

### 4.5. Statistical Analysis

Statistically significant differences between groups were determined using the Fisher Exact Test (GraphPad). A *p* value < 0.05 was considered statistically significant. Graphs were prepared using GraphPad Prism v. 9.5.1.

## Figures and Tables

**Figure 1 ijms-26-04120-f001:**
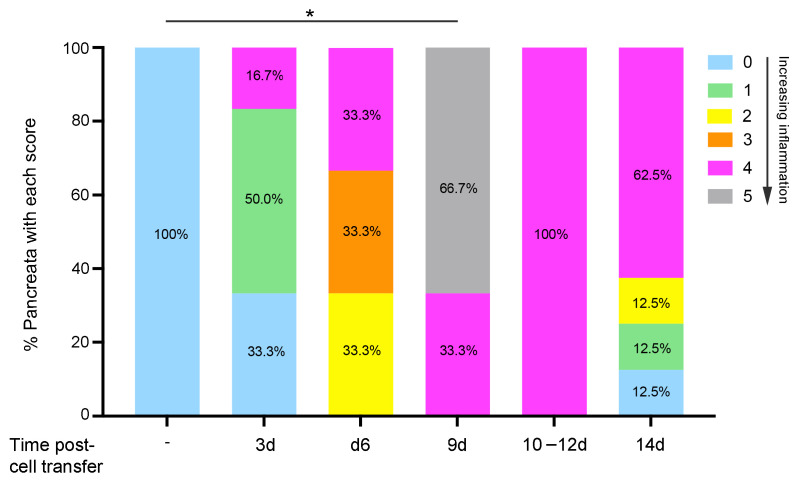
Intra-pancreatic cell infiltration in RIP-OVA^hi^ host mice peaks by day 9 after transfer of unlabeled or CFSE-labeled naïve OTI cells (2 × 10^6^) and activated OTII T cells (2 × 10^6^). Histological assessment of host RIP-OVA^hi^ pancreata shows the percentage of pancreata with each score for exocrine tissue infiltration and islet inflammation (Scores 0–5, as defined in score key and Section 4). Statistical analyses compared the percentage of pancreata with a given score between RIP-OVA^hi^ mice with and without the transfer of OTI + OTII cells and between groups of RIP-OVA^hi^ mice at different times after the transfer of OT cells. *n* = 3–8 mice/group except for RIP-OVA^hi^ mice without cell transfer, where *n* = 2 mice/group. * *p* = 0.04 (Score 0); Fisher’s exact test.

**Figure 2 ijms-26-04120-f002:**
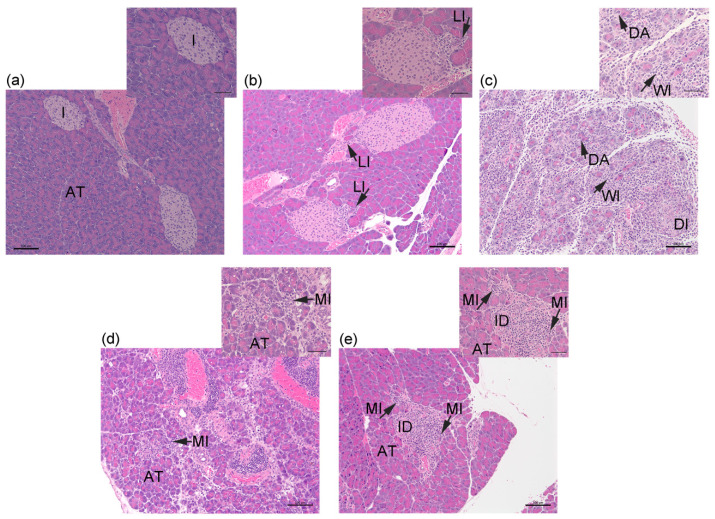
Histological appearance of RIP-OVA^hi^ pancreata at 3–14 days after transfer of CFSE-labeled or unlabeled OTI (2 × 10^6^) + OTII (2 × 10^6^) T cells: (**a**) Wildtype (WT) RIP-OVA^hi^ pancreas without OT cell transfer (Score 0) shows intact acinar tissue (AT) and intact islets (I). (**b**–**e**) Representative images at (**b**) 3 days, (**c**) 9 days), (**d**) 11 days, and (**e**) 14 days after OT cell transfer show a light cellular infiltrate (LI) near acinar tissue (Score 2) in (**b**); widespread immune cell infiltration (WI), damaged islet tissue (DI) and damaged acinar tissue (DA) with general disruption of the exocrine pancreas (Score 5) in (**c**); patches of mild immune cell infiltration (MI) within acinar tissue (Score 4) in (**d**,**e**) and islet destruction (ID) in (**e**). (**a**–**e**) Scale bar = 100 μm; scale bar for inset = 50 μm.

**Figure 3 ijms-26-04120-f003:**
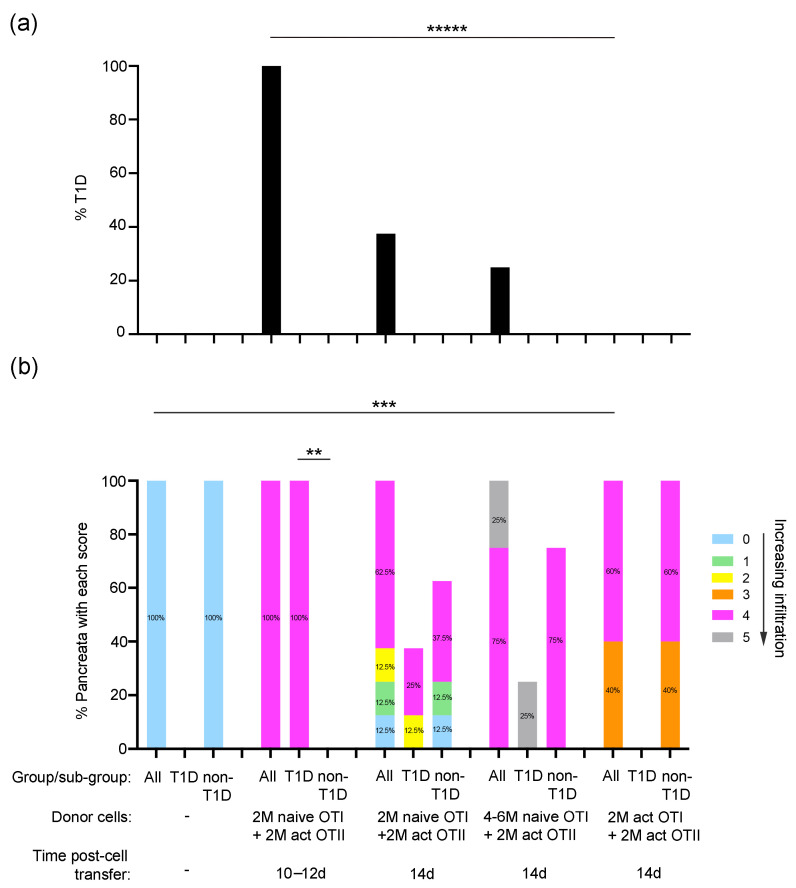
Pancreas infiltration and peak T1D incidence correlates with dose-dependent transfer of naïve OTI cells and activated OTII T cells in RIP-OVA^hi^ host mice: (**a**) Prevalence of diabetes in RIP-OVA^hi^ mice from (**b**) at 10–14 days post-cell transfer. ***** *p* = 0.008, Fisher’s exact test. (**b**) Histological assessment of host pancreata from RIP-OVA^hi^ mice with or without T1D shows the percentage of pancreata with each score for exocrine tissue infiltration and islet inflammation (Scores 0–5, as defined in score key and Section 4). Statistical analyses compare the percentage of pancreata with a given score between RIP-OVA^hi^ mice (All) with and without transfer of OTI + OTII cells (All) and between TID and non-T1D sub-groups. *n* = 4–8 mice/group except for WT RIP-OVA^hi^ mice without cell transfer, where *n* = 2 mice/group; ** *p* = 0.03 (Score 4), *** *p* = 0.02 (Score 0); Fisher’s exact test.

**Figure 4 ijms-26-04120-f004:**
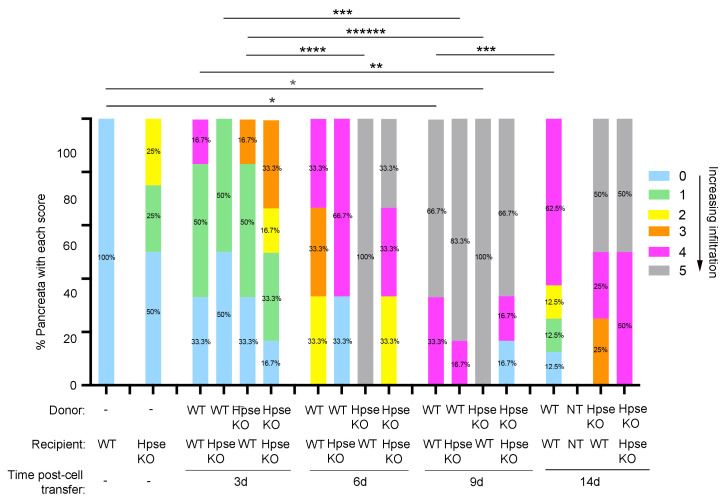
Histological assessment of pancreata from WT RIP-OVA^hi^ and HPSE-1-deficient (HPSE KO) RIP-OVA^hi^ host mice at 3 to 14 days after the adoptive transfer of unlabeled or CFSE-labeled WT OT (OTI + OTII) or HPSE-1-deficient (HPSE KO) OT T cells. Data show the percentage of pancreata with each score for exocrine tissue infiltration and islet inflammation (Scores 0–5, as shown in score key). *n* = 3–8 pancreata/group except for WT RIP-OVA^hi^ (no cell transfer), where *n* = 2 pancreata/group; * *p* = 0.04 (Score 0, WT host (no transfer) versus WT donor/WT recipient, 9 d; Score 5, WT host (no transfer) versus HPSE KO donor/WT recipient, 9 d), ** *p* = 0.03 (Score 4), *** *p* = 0.02 (Score 5, WT donor/WT recipient, 9 d versus 14 d; Score 5, WT donor/HPSE KO recipient, 3 d versus 9 d), **** *p* = 0.01 (Score 5), ****** *p* = 0.002 (Score 5); and Fisher’s exact test.

**Figure 5 ijms-26-04120-f005:**
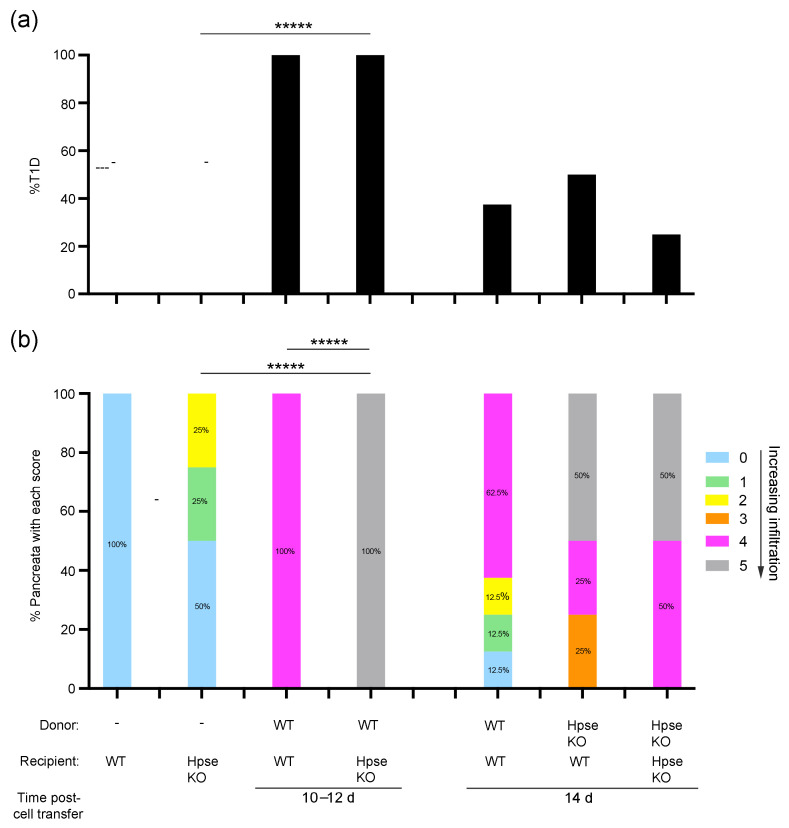
Exocrine tissue inflammation and diabetes induced by WT OT (OTI + OTII) or HPSE-1-deficient OT T cells at 10–14 days post-transfer to WT RIP-OVA^hi^ and HPSE-1-deficient RIP-OVA^hi^ host mice: (**a**) Incidence of diabetes in WT RIP-OVA^hi^ mice or HPSE-1-deficient (HPSE KO) RIP-OVA^hi^ mice from (**b**), at 10–14 days post-transfer of OT I and activated OTII T cells. *n* = 4–8 mice/group except for WT RIP-OVA^hi^ mice (no cell transfer), where *n* = 2 mice/group. ***** *p* = 0.008, Fisher’s exact test. (**b**) Histological assessment of host pancreata from (**a**) shows the percentage of pancreata with each score for exocrine tissue infiltration and islet inflammation (Scores 0–5, as defined in score key and Section 4). ***** *p* = 0.008 (Score 5), Fisher’s exact test.

**Figure 6 ijms-26-04120-f006:**
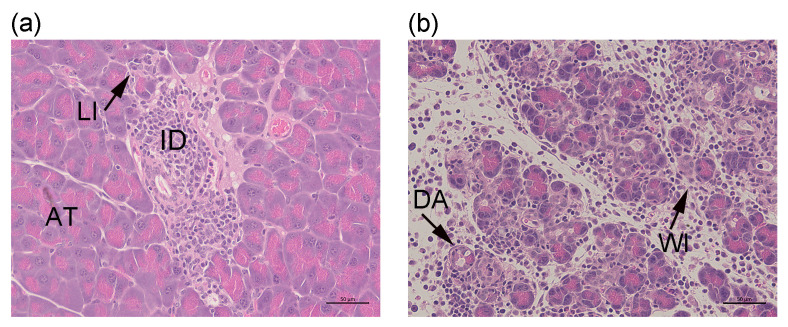
Histological appearance of WT or HPSE-1-deficient RIP-OVA^hi^ pancreata at 10–14 days after the transfer of WT OT (OTI + OTII) or HPSE-1-deficient OT (OTI + OTII) T cells. (**a**,**b**) Pancreata from (**a**) WT RIP-OVA^hi^ and (**b**) HPSE-1-deficient RIP-OVA^hi^ recipient mice with diabetes at 12 days after the transfer of WT OT T cells. Representative images show a light cellular infiltrate (LI) in acinar tissue (AT) and islet destruction (ID) (Score 4) in (**a**) and widespread immune cell infiltration (WI), damaged acinar tissue (DA), and the disruption of exocrine pancreas (Score 5) in (**b**). Scale bar = 50 μm.

## Data Availability

Data reported in this study are stored at the JCSMR and are freely available to other researchers upon request.

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
