# Peer review of "Transient Inflammation of Pancreatic Exocrine Tissue in Autoimmune Diabetes Follows Onset of Islet Damage and Utilizes Heparanase-1"

_ijms, 2025, doi:10.3390/ijms26094120_

Round 1

Reviewer 1 Report

Comments and Suggestions for Authors

The manuscript of Charmaine J. Simeonovic et al., presents a comprehensive histological and immunological analysis of the role of donor and recipient heparanase-1 (HPSE-1) in the development of pancreatic inflammation and Type 1 diabetes (T1D) in RIP-OVAhi mice. The study is methodologically robust, addresses a key issue in autoimmune diabetes research, and includes well-organized figures and accurate histological data. The findings offer valuable insights into the differential roles of donor and host HPSE-1 in both the endocrine and exocrine compartments of the pancreas.

The work is interesting, I think the authors should measure heparanase activity using an ELISA test or WB or other techniques to quantify the products or the he reduction of a concentration of heparan or heparansulfate to establish a more direct correlation between enzyme activity and disease progression.

Acronyms like LI, MI, DI, DA, ID appear without immediate expansion. Define each at first use in both text and figure legends.

Ensure uniform reporting of p-values and tests used

The work would also be enriched by the inclusion of a reference of an experimental study of September 2021 on the anti-inflammatory effects of heparanase inhibitors (such as the reduction of tissue factor).

Author Response

Please see the attachment (Reviewer 1_ijms-3579798_Simeonovic et al.pdf)

Reviewer 2 Report

Comments and Suggestions for Authors

In this manuscript, the authors use the RIP-OVA mouse model and OVA-specific T cell transfer system to demonstrate the following: (1) Beta-cell-specific T cells induce both islet and exocrine tissue inflammation. However, exocrine inflammation is more transient, occurs later, and is less severe compared to islet inflammation and injury. (2) Host-derived HPSE-1 contributes to inflammation in both the exocrine and endocrine compartments of the pancreas, whereas donor T cell-derived HPSE-1 primarily facilitates islet infiltration and beta-cell destruction. Overall, the authors build upon their previous study, showing that immune cells and HPSE-1 contribute to inflammation in both the exocrine and endocrine compartments of the pancreas during T1D development. However, exocrine pancreas injury is not required for the initiation or persistence of antigen-specific islet damage or for diabetes onset.

1. The study uses a well-established model to investigate the relationship between islet inflammation, exocrine tissue inflammation, T1D, and HPSE-1. Can the authors provide human data to validate these findings?

2. The authors demonstrate that HPSE-1 plays an important role in both donor and recipient functions, which is logical given its effects on the extracellular matrix. Since heparan sulfate and chondroitin sulfate are both glycosaminoglycans crucial for ECM integrity, it would be interesting to explore whether chondroitin sulfate also plays a significant role in these processes. The authors may discuss this possibility or provide supporting evidence if desired.

3. The authors should label p-values directly in the figures or use a standardized method for reporting p-values, rather than using multiple symbols ( * and #), which may be confusing to readers.

Author Response

Please see the attachment (Reviewer 2_ijms-3579798_Simeonovic et al.pdf)

Round 2

Reviewer 1 Report

Comments and Suggestions for Authors

The manuscript is worthy of publication.

I would appreciate if the authors would consider including in their list of references an experimental study of 2021  on effect of heparanase inhibitor on platelets and endothelial cells, published in the Journal of Thrombosis and Haemostasis.

Author Response

Please see uploaded file: Reviewer 1_Round 2 response_ijms-3579798_Simeonovic et al.pdf

Reviewer 2 Report

Comments and Suggestions for Authors

The authors have addressed my questions.

Author Response

Please see uploaded file: Reviewer 2_Round 2_ijms-3579798_Simeonovic et al.pdf
